# Small molecule modulation of the *Drosophila* Slo channel elucidated by cryo-EM

Tobias Raisch [1], Andreas Brockmann [2,3], Ulrich Ebbinghaus-Kintscher[2], Jörg Freigang [2], Oliver Gutbrod[2], Jan Kubicek[4], Barbara Maertens[4], Oliver Hofnagel[1] & Stefan Raunser [1]✉

Slowpoke (Slo) potassium channels display extraordinarily high conductance, are synergistically activated by a positive transmembrane potential and high intracellular $Ca^{2+}$ concentrations and are important targets for insecticides and antiparasitic drugs. However, it is unknown how these compounds modulate ion translocation and whether there are insect-specific binding pockets. Here, we report structures of *Drosophila* Slo in the $Ca^{2+}$-bound and $Ca^{2+}$-free form and in complex with the fungal neurotoxin verruculogen and the anthelmintic drug emodepside. Whereas the architecture and gating mechanism of Slo channels are conserved, potential insect-specific binding pockets exist. Verruculogen inhibits $K^+$ transport by blocking the $Ca^{2+}$-induced activation signal and precludes $K^+$ from entering the selectivity filter. Emodepside decreases the conductance by suboptimal $K^+$ coordination and uncouples ion gating from $Ca^{2+}$ and voltage sensing. Our results expand the mechanistic understanding of Slo regulation and lay the foundation for the rational design of regulators of Slo and other voltage-gated ion channels.

[1] Department of Structural Biochemistry, Max Planck Institute of Molecular Physiology, 44227 Dortmund, Germany. [2] Bayer AG, Crop Science Division, D-40789 Monheim am Rhein, Germany. [3] Rheinische Friedrich-Wilhelms-Universität Bonn, D-53113 Bonn, Germany. [4] Cube Biotech GmbH, Alfred-Nobelstr. 10, 40789 Monheim, Germany. ✉email: stefan.raunser@mpi-dortmund.mpg.de

*D*rosophila Slowpoke (Slo) is the founding member of a class of homotetrameric ion channels that are characterized by their high selectivity for potassium ions and their extraordinarily high single-channel conductance[1–4]. These BK ("big potassium") channels are synergistically activated by high intracellular $Ca^{2+}$ concentrations and a positive transmembrane potential[5–7]. They are ubiquitously expressed in metazoa and take part in a wide range of physiological functions including $K^+$ release in the kidney and audition[1,2,8,9]. Of special importance is their role in neuronal and muscle tissues where the $K^+$ concentration needs to be coupled to intracellular $Ca^{2+}$ concentration and the transmembrane potential. Here, BK channels crucially contribute to neurotransmitter release, repolarization of action potentials, and smooth muscle tension[10–13].

$K^+$ ions are translocated through a pore domain comprising a wider aqueous central cavity that spans the inner leaflet, and a narrow selectivity filter that spans the outer leaflet of the plasma membrane and can accommodate four $K^+$ ions[14–17]. The pore domain is surrounded by four voltage sensor domains (VSDs) that move within the membrane according to the transmembrane potential, thereby inducing a conformational change in the central cavity that changes its accessibility for ions[15–21]. In addition to this transmembrane domain (TMD), Slo and its homologs feature a large, domain-swapped intracellular gating ring, which senses intracellular $Ca^{2+}$ concentrations. The binding of $Ca^{2+}$ to a total of eight binding sites leads to an expansion of the gating ring[22–24]. As it is directly connected to the pore domain via a short linker, this conformational change also causes the TMD S6 helix lining the central cavity to adopt a different, active conformation which allows easy access of $K^+$ to the selectivity filter[15–17,25].

Ion channels are amongst the most important targets for neuroactive insecticides[26,27]. Due to their crucial role in neuronal signal transduction and muscle contraction in nematodes and insects, Slo channels are a prime target for such neurotoxic insecticides and pharmacological substances[28]. Multiple natural substances confer neurotoxicity by blocking Slo. These include peptide toxins of spider and scorpion venoms, some of which specifically inhibit insect but not vertebrate Slo channels[28,29]. On the other hand, a class of low molecular weight fungal toxins display rather broad species specificity and can inhibit invertebrate and also mammalian Slo[28,30,31]. Despite the importance of Slo channels as insecticide targets, little is known about the structural and mechanistic details explaining how these proteins can be modulated by small molecule inhibitors and activators. However, such information would be essential for structure-guided optimization of existing compounds toward improved species specificity as well as the rational design of novel molecules targeting insect Slo channels.

Here, we report four cryo-EM structures of the prototype insect Slo channel from *Drosophila melanogaster* in the $Ca^{2+}$-bound and $Ca^{2+}$-free conformations and in complex with the ligands verruculogen and emodepside. We identify insect-specific drug binding pockets and, based on structural analyses and patch-clamp assays, we provide the mechanistic basis of how small molecule ligands can modulate the activity of Slo channels in very different ways by binding to overlapping sites in the central cavity, and are thus a major step toward the rational design of novel Slo modulators for medical, veterinary and agricultural use.

## Results

**Conserved $Ca^{2+}$-dependent gating mechanism.** Insecticides and nematicides should be species-specific in order to avoid interference with Slo channels of vertebrates or non-pest insects like bees. Potential insect-specific drug-binding pockets need to be large enough to accommodate small molecules, be solvent-accessible, and should not exist in vertebrate Slo channels. Importantly, they need to block or deregulate $K^+$ translocation. The most direct inhibitory sites in ion channels are the entryways to the selectivity filter (Supplementary Fig. 3A, B), as a physical block would restrict or completely prevent ion translocation. Indeed, different classes of ion channels can be inhibited by small molecules sterically blocking the intracellular entryway[32–35], while a class of peptidic spider and scorpion toxins plug the selectivity filter of Slo channels from the extracellular side[29,36]. However, as the extracellular site is very shallow, it might not represent an ideal binding site for low molecular weight compounds, and the high conservation of the intracellular site would pose challenges regarding species specificity.

To computationally identify locations of novel potential drug-binding pockets in insect Slo channels, high-resolution structures of these channels are needed. Therefore, we heterologously expressed and purified Slo from *Drosophila melanogaster* (*Dm*; Methods, Fig. 1A, Supplementary Fig. 1) and determined cryo-EM structures of the channel in the $Ca^{2+}$-bound and $Ca^{2+}$-free state. The high resolution of the reconstructions, 2.4 and 2.7 Å, respectively, allowed us to build atomic models of the channel with high accuracy also on the side chain level, which is important for in-silico drug screening (Methods, Fig. 1B–E, Supplementary Fig. 2A, B, Supplementary Movie 1). Expectedly, the structures are very similar to those of the human and *Aplysia* homologs[15–17], confirming that the overall architecture of the channel and the gating mechanism are conserved between metazoans (Supplementary Fig. 2C–H).

We then analyzed the surfaces of *Drosophila* and human Slo for potential binding pockets using the program BiteNet (https://sites.skoltech.ru/imolecule/tools/bitenet/), which uses an object detection convolutional neural network to predict binding sites in protein structures[37]. The program identified several solvent-accessible binding pockets in the gating ring and central cavity of both channels (Fig. 2A–D, Supplementary Movie 2). Importantly, all of them differ in their conformation between the $Ca^+$-bound or the $Ca^+$-free states and have thus the potential to lock the channel in either the open or the closed conformation, thereby uncoupling $K^+$ translocation from regulation by intracellular $Ca^{2+}$ levels and the transmembrane potential with severe and potentially lethal physiological consequences.

The most promising binding pocket which was only identified in *Drosophila* Slo is located on the RCK2 domain with additional contributions from RCK1. This pocket, termed RCK2 pocket, changes its shape upon the $Ca^{2+}$-induced rearrangement of the gating ring (Fig. 3A, B, Supplementary Movie 2). Thus, occupying it with a small molecule tailored to fit one of the states might stabilize the gating ring in that state. Importantly, one side of the pocket is lined with non-conserved amino acids, which differ between *Drosophila* Slo and human Slo1 (Fig. 3A, B, Supplementary Fig. 1, Supplementary Fig. 4A). Therefore, this pocket offers the possibility of designing insect-specific Slo modulators. Furthermore, as several amino acids lining this pocket are not conserved between vertebrates and helminths (Supplementary Fig. 4A), the RCK2 pocket might present the chance of developing novel specific anthelmintics.

Besides this unique binding pocket that was only identified in *Drosophila* Slo, BiteNet identified sites that are either unique for human Slo1 or exist in both channels. Two potential binding sites in the gating ring were only identified in the $Ca^{2+}$-bound active state as they are lined by residues of the moving RCK1-lobe on the one side, and either residues of the same subunit or the respective neighboring subunit (Fig. 2A). These pockets, termed RCK1 pocket A and B, are mostly composed of highly conserved

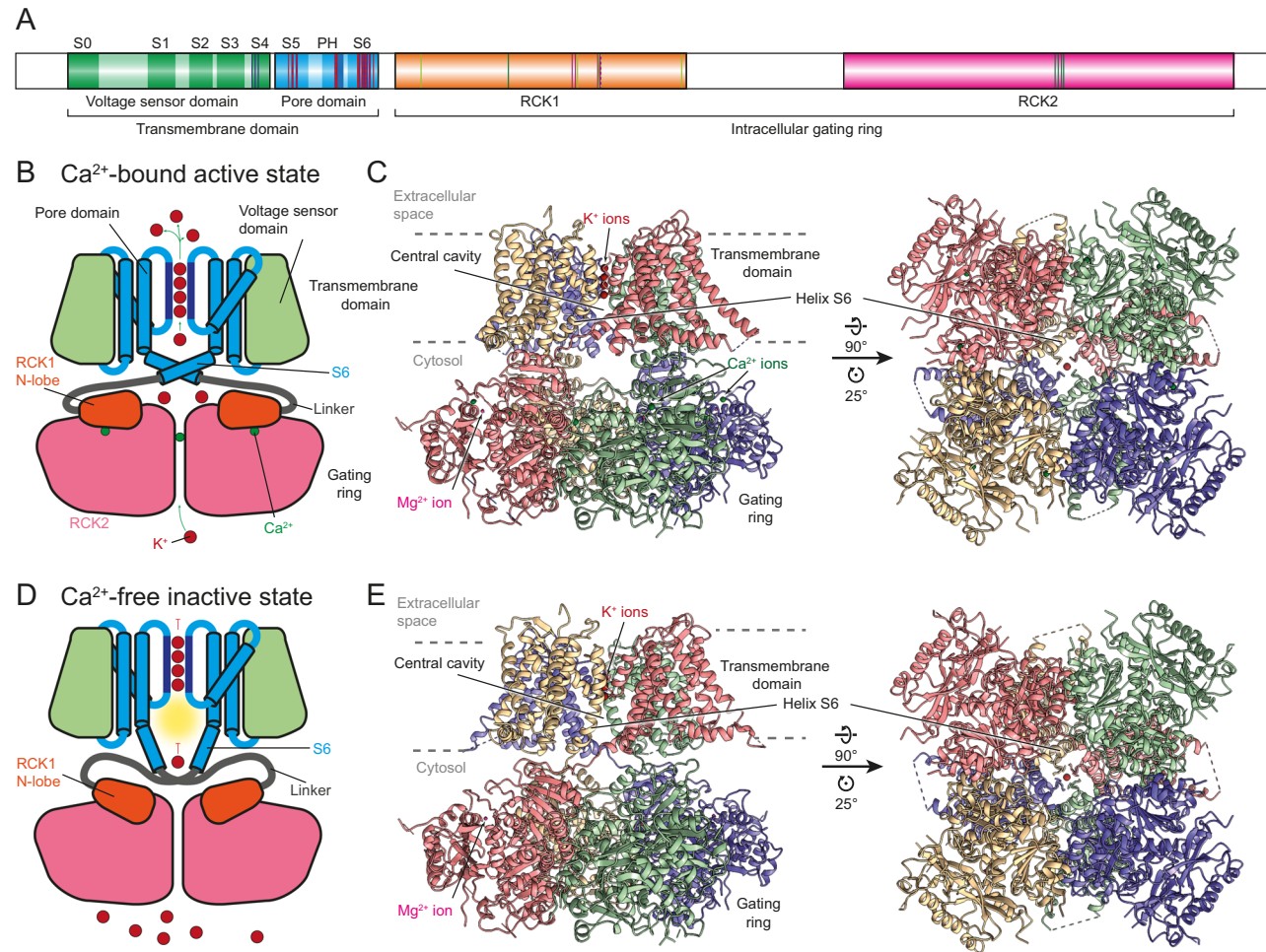

**Fig. 1 *Drosophila* Slo features all conserved hallmarks of Ca$^{2+}$-gated K$^+$ channels. A** Schematic representation of Slo. The protein comprises an N-terminal TMD consisting of a voltage sensor domain (VSD; green) and a pore domain (PD; blue) that assembles around the central ion conducting channel, and two C-terminal regulatory domains for K$^+$ conductance (RCK1 and RCK2; orange and pink, respectively) that constitute the intracellular gating ring. Transmembrane helices in the VSD and the PD are highlighted and labeled. The selectivity filter C-terminal of helix PH in the PD is highlighted in darker blue. The three gating charge arginine residues in helix S4 of the VSD are indicated in dark blue. Residues interacting with the two Ca$^{2+}$ ions in the gating ring are marked in light and dark green, respectively, and residues involved in Mg$^{2+}$ binding in pink. Residues lining the verruculogen binding site in the PD are marked in red, and residues interacting with emodepside in purple. **B** Schematic representation of Slo in the Ca$^{2+}$-bound active conformation. Ca$^{2+}$ ion binding to the gating ring locks the RCK1 N-lobes in an outward-facing conformation. This rearrangement is transferred into the pore domain via the linker between helix S6 and the RCK1 N-lobe, and the C-terminal end of S6 adopts a kinked conformation. As a consequence, the entry to the central cavity is wide and hydrophobic patches along its walls are buried and K$^+$ ions can access the selectivity filter and be translocated to the extracellular space. **C** Structure of *Drosophila* Slo in the Ca$^{2+}$-bound active conformation in two orientations with colors assigned by macromolecular chain to visualize the domain swap between TMD and gating ring. **D** Schematic representation of Slo in the Ca$^{2+}$-free inactive conformation. Helix S6 of the pore domain adopts a straight conformation that exposes hydrophobic patches along the constricted cavity walls which repulse K$^+$ ions. **E** Structure of *Drosophila* Slo in the Ca$^{2+}$-free inactive conformation in the same two orientations as in panel **C**.

key residues (Supplementary Figs. 1, 3E–H, and 4B, C). Hence, while they have the potential to be utilized to lock the channel in the open state by tight binding of a small molecule, this would most likely happen without high species specificity.

In the Ca$^{2+}$-free state, a deep, hydrophobic pocket is located below helix S6 in the cavity close to the selectivity filter in human and *Drosophila* Slo (Figs. 2B, D and 3C, D, Supplementary Figs. 3C, D and 4D, Supplementary Movie 2). Since this pocket, termed S6 pocket, differs only slightly between insect and vertebrate Slo homologs and the non-conserved residues cluster in the back of the pocket where it opens toward the lipid bilayer (Supplementary Figs. 1, 3D, and 4D), it might be difficult to find or create compounds with high species specificity. However, the S6 pocket fulfills all requirements of a druggable pocket since it is deep, mostly hydrophobic, solvent-accessible, and a small

molecule bound to it might lock the channel in the closed conformation.

**Verruculogen restricts access to the selectivity filter.** To elucidate whether known Slo ligands bind to the predicted binding pockets and how they modulate channel activity, we examined the complexes of Slo with the tremorgenic mycotoxin verruculogen[30,38,39] and the anthelmintic drug emodepside[40,41], respectively. Verruculogen belongs to a class of natural Slo inhibitors that contain a common indole core structure. It inhibits Slo with an IC$_{50}$ in the low nM range (Supplementary Fig. 5A)[38]. Emodepside is a semi-synthetic cyclo-octadepsipeptide[40,41]. On the molecular level, emodepside activates latrophilins[42] and, more importantly, deregulates Slo in motor neurons[43,44]. In presence of

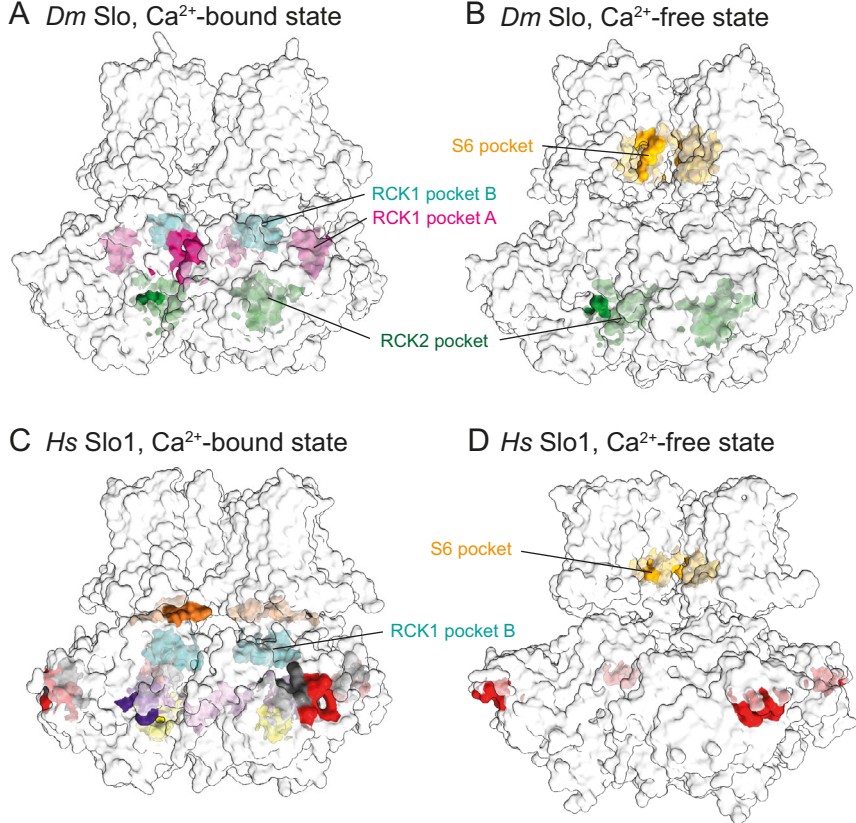

**Fig. 2 Identification of potential drug-binding pockets on *Drosophila* Slo. A**, **B** Location of potential drug-binding pockets on the Ca$^{2+}$-bound (panel **A**) and Ca$^{2+}$-free (panel **B**) conformations of *Drosophila* Slo. BiteNet[37] identified two sites in the RCK1 domain which are defined by residues of the neighboring (pocket A, pink) or the same (pocket B, cyan) subunit and were only identified in the Ca$^{2+}$-bound conformation. A third pocket is present in the RCK2 domain in both conformations (green). Furthermore, an S6 pocket specific to the closed state is present in the cavity within the TM domain (yellow). **C**, **D** Location of potential drug-binding pockets on the Ca$^{2+}$-bound active (panel **C**; PDB 6V38) and the Ca$^{2+}$-free inactive (panel **D**; PDB 6V3G) conformations of human Slo1. In addition to the RCK1 pocket A and the S6 pocket that have also been identified in *Drosophila* Slo, several additional pockets have been identified in the gating ring (highlighted in red, gray, purple, violet, and yellow) and along the linker connecting TMD and gating ring (brown).

emodepside, which affects Slo with an IC$_{50}$ in the low μM range (Supplementary Fig. 6A), *Drosophila* Slo is insensitive to the regulation by Ca$^{2+}$ concentrations and transmembrane voltage and its ion conducting behavior resembles that of a non-regulated K$^+$ pore (Supplementary Fig. 6B)[38].

We determined cryo-EM structures of verruculogen-bound and emodepside-bound Slo in presence of Ca$^{2+}$ at resolutions of 2.7 Å and 2.6 Å, respectively (Figs. 4 and 5). The verruculogen-bound Slo structure reveals that the compound binds in a four-fold symmetric manner in the S6 pocket which is one of the binding sites predicted by our BiteNet analysis (Figs. 2B, 3D and 4A, B, Supplementary Fig. 5B, Supplementary Movie S3). Thus, verruculogen binding appears to require the Ca$^{2+}$-free state where helix S6 and the S6-RCK1 linker are in their closed conformation, analogously to what has been observed for the toxin paxilline which might bind to the same site[45,46]. Conversely, verruculogen locks helix S6 and thereby the pore domain in the closed conformation. In our structure, strikingly, the gating ring adopts the Ca$^{2+}$-bound open conformation (Supplementary Fig. 5C) while S6 is in the Ca$^{2+}$-free closed conformation (Fig. 4C). This discrepancy is compensated by a slightly stronger kink of S6 around I333 and a more extended conformation of the following linker (residues 341–354) connecting to the gating ring (Fig. 4C).

Verruculogen binds to Slo in a mostly hydrophobic manner and largely by perfect shape-complementarity to the binding pocket that is lined by residues L253, I256, F257, and V260 of

helix S5, the gatekeeper T301 and its neighbor S300, and helix S6 residues F321, L322, G325, L326, I328, F329, C332, and I336 (Fig. 4D, Supplementary Fig. 1). As almost all residues lining the pocket are highly conserved between animals (Supplementary Figs. 1 and 4D), it is not surprising that verruculogen shows no pronounced species selectivity and can inhibit insect Slo[38] and mammalian homologs[30].

One key aspect of why verruculogen is such a potent Slo inhibitor is the aliphatic isobutylene moiety that reaches into the central cavity (Fig. 4B, D, Supplementary Fig. 5D, E). We reasoned that this group might change the volume and the chemical environment in the central cavity and that this might prevent K$^+$ ions from efficiently reaching the selectivity filter and thereby the observed inhibition of ion conductance by verruculogen. While verruculogen did not induce any changes in the selectivity filter, the pore diameter in the central cavity is strongly reduced by the inactive conformation of helix S6 and resembles the inactive apo state, and verruculogen itself leads to a further constriction of up to 2 Å in the selectivity filter-adjacent part of the central cavity (Fig. 4E) Verruculogen also induces an increase in hydrophobicity as the more hydrophobic face of S6 points to the interior of the central cavity, and the aliphatic portion of verruculogen itself enhances this effect (Fig. 4F).

The constriction and increased hydrophobicity of the central cavity might cause a situation where K$^+$ ions are prevented from reaching the selectivity filter. Indeed, a superposition of our structure of verruculogen-bound Slo with a structure of the

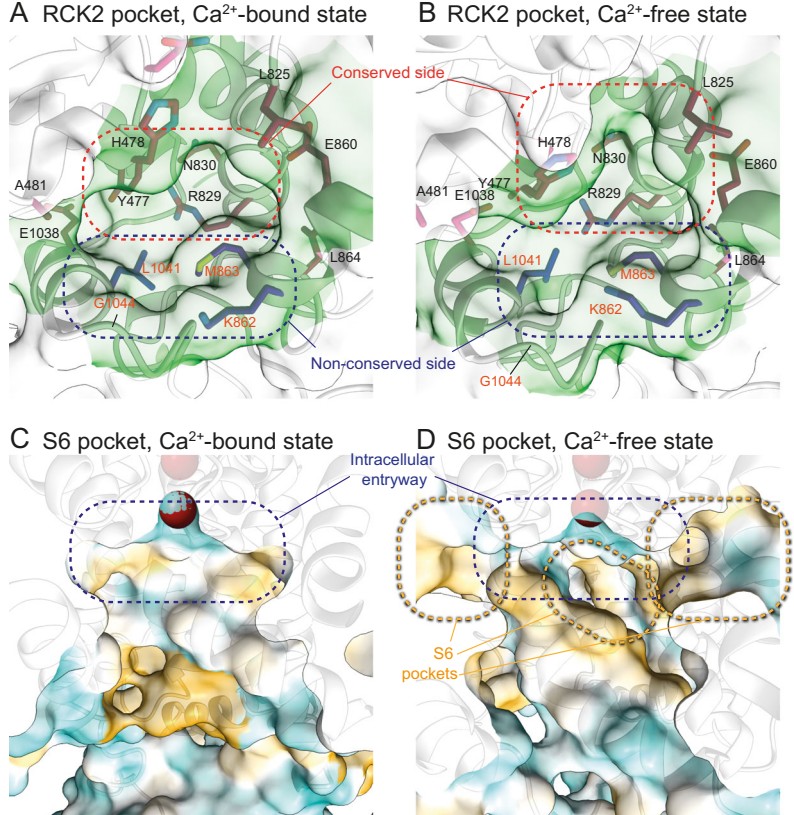

**Fig. 3 RCK2 and S6 pockets display characteristics of druggable pockets. A** Close-up view of the RCK2 pocket in the $Ca^{2+}$-bound active conformation. Amino acid side chains that line the pocket are shown in stick representation and colored according to conservation from blue (low conservation) to red (high conservation). Residues that are identical between *Drosophila* Slo and human Slo1 are labeled with black labels, residues that differ with orange labels. Intriguingly, the side of the pocket lined by K862, M863, L1041, and G1044 is not conserved and also differs between *Drosophila* and human. **B** Close-up view of the RCK2 pocket in the $Ca^{2+}$-free conformation. The shape of the pocket changes upon $Ca^{2+}$-induced rearrangement of the gating ring. **C** Surface representation of the central cavity in the $Ca^{2+}$-bound active conformation colored in a gradient from cyan over white to yellow according to increasing molecular lipophilicity potential (MLP). Half of the Slo tetramer was omitted to allow an unobstructed view into the internal pore. The intracellular entryway to the selectivity filter is indicated in blue. **D** Surface representation of the cavity in the $Ca^{2+}$-free inactive conformation. The rearrangement of helix S6 exposes hydrophobic pockets (indicated in orange) in the vicinity of the intracellular entryway.

bacterial channel KscA that was obtained at a high $K^+$ concentration[47] revealed that the Slo isobutylene moieties would be very close to a hydrated $K^+$ ion in the 'S6' position, i.e., the position inside the cavity where ions are focused before entering the selectivity filter (Supplementary Fig. 5F, G). While the space between the four isobutylene tails is just large enough for hydrated ions to pass through, their hydrophobic nature would make such an arrangement unfavorable. This indicates that verruculogen creates an efficient barrier preventing hydrated $K^+$ ions from reaching the proximity of the selectivity filter, thus leading to a complete block of Slo ion transduction without sterically plugging the pore.

**Emodepside prevents $Ca^{2+}$ and voltage sensing.** The emodepside-bound Slo structure revealed a conformation of Slo which is almost identical to the $Ca^{2+}$-bound apo state (Supplementary Fig. 6C). Emodepside binds across the central symmetry axis to the base of the central cavity and directly to the intracellular entryway of the selectivity filter (Fig. 5A–C, Supplementary Fig. 6D, Supplementary Movie 4). It thus binds near the position of verruculogen binding, namely close to the predicted S6 pocket. However, in contrast to verruculogen, it does not enter the pockets which are closed in this conformation (Supplementary Fig. 8D). The core of the molecule consists of an almost

fourfold-symmetric ring of four copies of N-methyl-L-leucine and four copies of D-lactate[41] and binds directly on top of the turns between the PH helices and the selectivity filter. Its four isobutyl side chains point outward and slightly upward in the direction of the clefts between S6 helices of two neighboring subunits and might help stabilize the open S6 conformation (Fig. 5C, D). In addition to this C4-symmetric core, two of the D-lactate copies are modified by large morpholinylphenyl groups that reach far into the hydrophobic pockets between helices PH and S6 and help anchor the compound in its binding site (Fig. 5C). As a consequence of emodepside being overall C2-symmetric, also the S6 helices of which two were slightly displaced from their active conformation by the two morpholino groups of emodepside deviate from the C4 symmetry (Fig. 5B, C).

The emodepside ring is amphipathic, and while the hydrophobic outer side faces the central cavity, all eight carbonyls point toward the center of the pore (Fig. 5C, D, Supplementary Fig. 6E). As emodepside spans the whole width of the cavity (Fig. 5F, Supplementary Fig. 6F), $K^+$ ions need to pass through this ring in order to reach the selectivity filter and be translocated. This is assisted by the carbonyl oxygens of the emodepside which, similar to the carbonyls of the selectivity filter, might help in focusing the arriving $K^+$ in the central S6 position below the emodepside, and hand them to the S5 position between the emodepside and the $K^+$ in position S4 of the filter. Indeed, we observe weak ordered

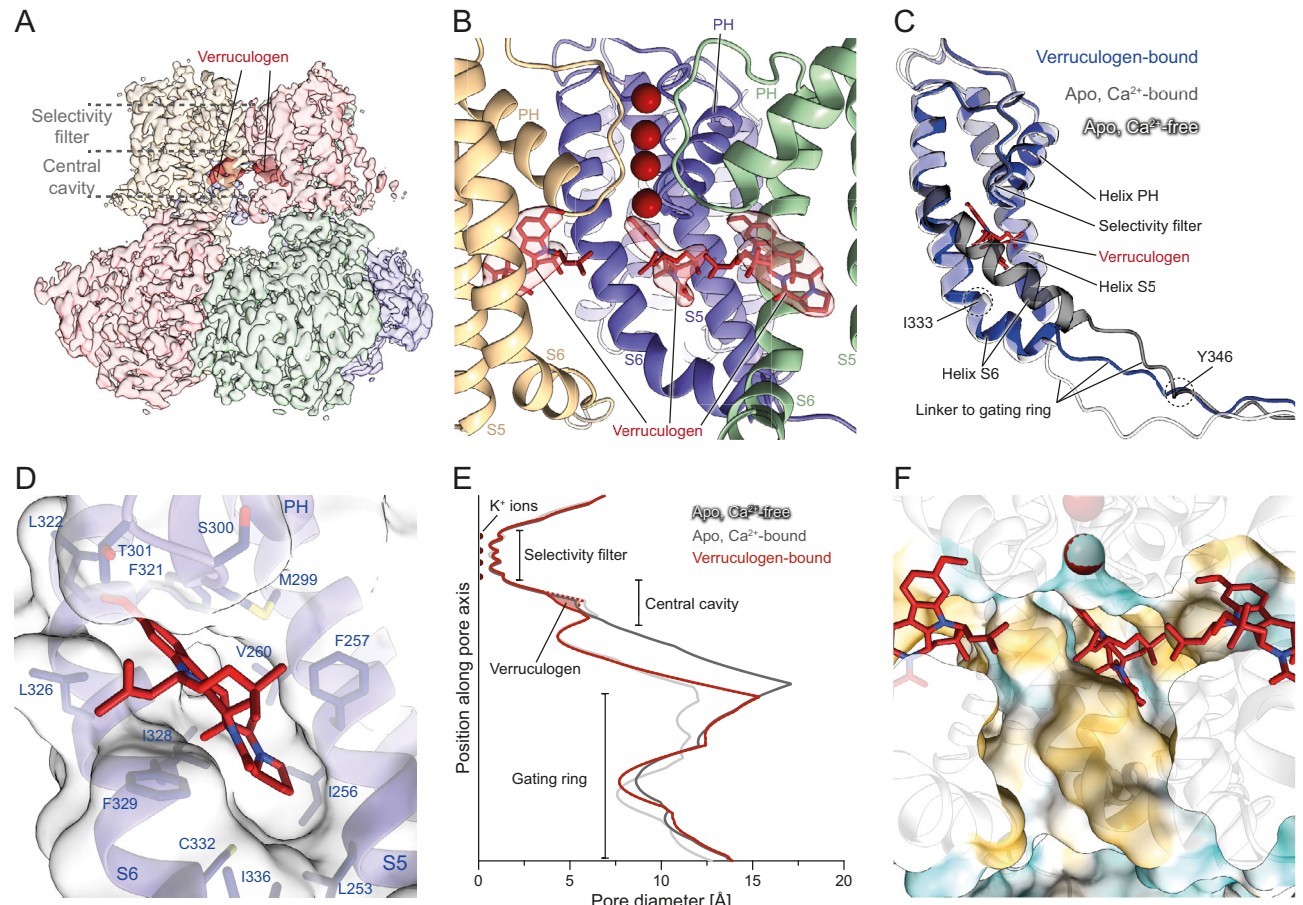

**Fig. 4 Verruculogen restricts access to the ion-conducting pore. A** Cryo-EM reconstruction of verruculogen-bound Slo colored by the macromolecular chain. Verruculogen molecules are shown as dark red surfaces. The position of the cavity and the selectivity filter are indicated by dashed lines. **B** View of the verruculogen-bound cavity shown from the side. One of the four macromolecular chains was omitted to allow an unobstructed view into the internal pore. The corresponding verruculogen molecules (red) are shown as sticks and are surrounded by the experimental Coulomb potential map (faint red). Verruculogen binds into a pocket between helices S5 and S6 and the turn between the PH helix and the selectivity filter. **C** Superposition of the hinge helix S6 conformation in the verruculogen structure (blue) with both apo structures (gray). When verruculogen is bound, helix S6 follows a trajectory that is almost identical to the $Ca^{2+}$-free conformation since the position of the $Ca^{2+}$-bound conformation would clash with verruculogen. This is compensated by a stronger kink around residue I333 and a more extended conformation of the linker which joins the $Ca^{2+}$-bound conformation around Y346. **D** Closeup view of the verruculogen binding site showing the exact fit of verruculogen into the pocket. The cavity is lined predominantly by hydrophobic residues of helices S5 and S6 as well as the turn C-terminal of helix PH. **E** Pore diameter plot of the verruculogen-bound conformation (red) as well as both apo states (gray). The red dashed line indicates the pore diameter of the verruculogen-bound conformation with verruculogen omitted from the structure, and the light red area the contribution of verruculogen to the narrower pore. Verruculogen induces a $Ca^{2+}$-free-like diameter of the cavity. **F** Surface representation of the pore around the verruculogen-bound cavity colored in a gradient from cyan over white to yellow according to increasing molecular lipophilicity potential (MLP). Half of the Slo tetramer was omitted to allow an unobstructed view into the internal pore. The exposed hydrophobic face of helix S6 appears even more dominant and, along with the aliphatic tail of verruculogen pointing into the cavity, results in an overall very hydrophobic environment.

density which we have modeled as water molecules at the S5 and S6 positions (Fig. 5D, Supplementary Fig. 6G), and it is conceivable that $K^+$ could be accommodated there. However, the coordination appears suboptimal since opposite oxygen atoms in the emodepside are slightly further apart compared to those in the selectivity filter (~5.5–6 Å compared to ~4.5 Å; Supplementary Fig. 6E) and, therefore, also the pore is slightly wider (Fig. 5E). This suboptimal $K^+$ coordination combined with emodepside occupying a position where ions are usually focused in the S6 position by the PH helix dipoles[14,47,48] might explain the observed lower conductance of Slo under high $Ca^{2+}$ conditions when emodepside is present[38].

This hypothesis of suboptimal ion coordination appears likely in light of a superposition of emodepside-bound Slo with KscA in complex with a hydrated $K^+$ ion in the S6 position[47]. As this hydrated $K^+$ ion would indeed be located exactly where we observe the weak density we modeled as water (Supplementary Fig. 7A–C), it is conceivable that the carbonyls of emodepside might indeed be able to coordinate a dehydrated $K^+$ ion. However, close inspection reveals that such coordination might be imperfect since at least in the conformation emodepside adopts in our structure, the coordination angles are slightly distorted when compared with the perfect hydration shell of the S6 $K^+$ in KscA (Supplementary Fig. 7C, D). Such imperfect coordination could possibly explain the lower currents of activated Slo in the presence of emodepside. In this context, it is interesting to mention a class of negatively charged activators (NCAs) that have been proposed to enhance $K^+$ translocation of

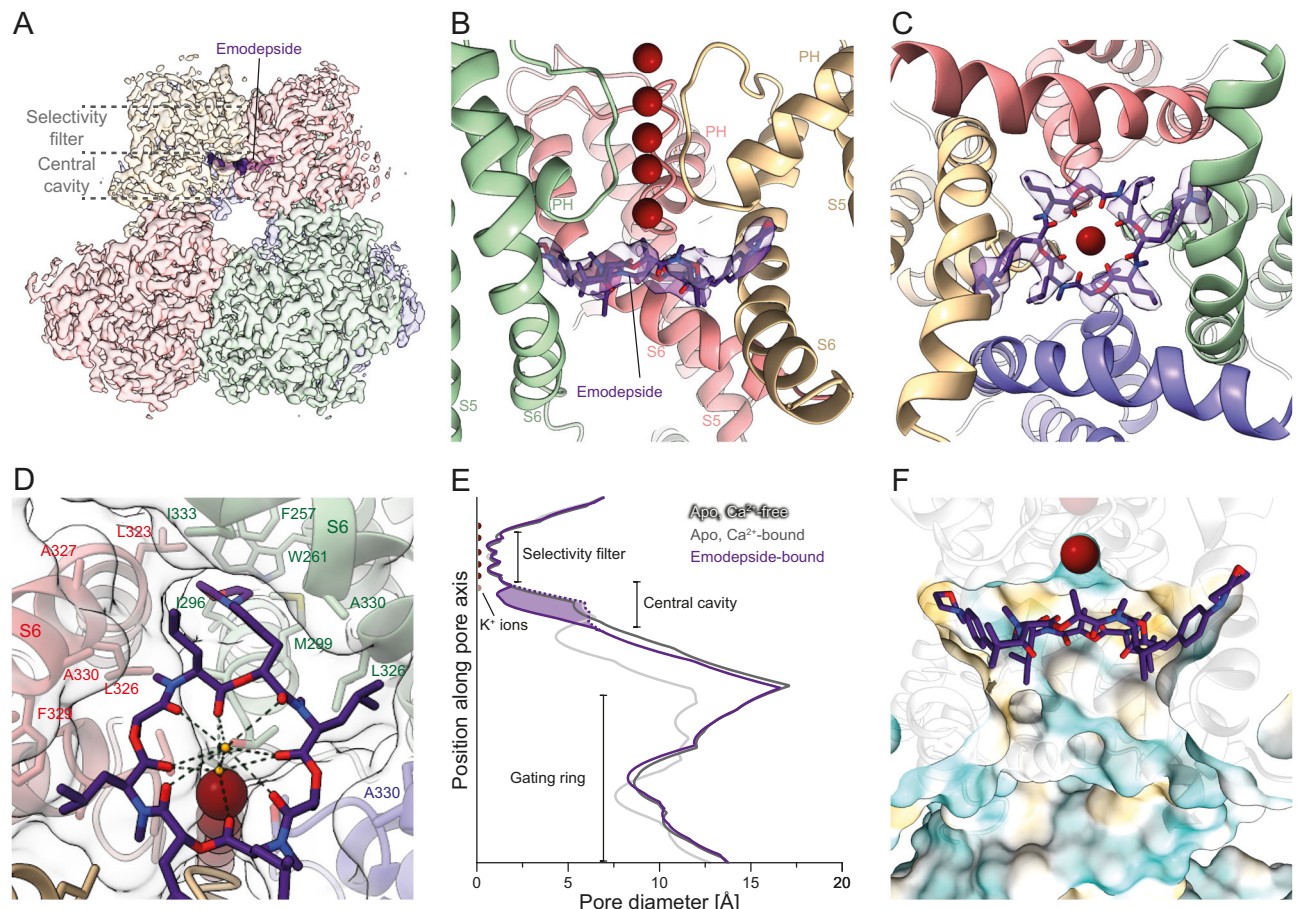

**Fig. 5 Emodepside acts as an additional selectivity filter layer. A** Cryo-EM reconstruction of emodepside-bound Slo colored by the macromolecular chain. Emodepside is shown as a dark purple surface. **B** View of the emodepside-bound cavity shown from the side. One of the four macromolecular chains was omitted to allow an unobstructed view into the internal pore. Emodepside is shown as sticks and is surrounded by the experimental Coulomb potential map (faint purple). **C** View from the cytosol into the emodepside-bound cavity illustrating that the central opening in the emodepside is wide enough to translocate K$^+$ ions. **D** Close-up of the emodepside binding site showing how the molecule fits on the intracellular entryway and the morpholino and isobutyl side chains reach into pockets between the S6 helices of Slo. Two water molecules that are stabilized by emodepside are shown as small red spheres, and hydrogen bonds between the emodepside and the waters are indicated by dashed lines. **E** Pore diameter plot of the emodepside-bound conformation (purple) as well as both apo states (gray). The pore is only slightly wider at the emodepside position than at the selectivity filter. The purple dashed line indicates the pore diameter of the emodepside-bound conformation with emodepside omitted from the structure, and the light purple area the contribution of emodepside to the narrower pore. Apart from the constriction caused by the emodepside itself, the pore width of the emodepside-bound conformation of Slo is very similar to the Ca$^{2+}$-bound apo state. **F** Cut-open surface representation of the pore around the emodepside-bound cavity colored in a gradient from cyan over white to yellow according to increasing molecular lipophilicity potential (MLP). The cavity appears to be similarly wide and hydrophilic as in the Ca$^{2+}$-bound apo state (Fig. 3C), and less hydrophobic compared to the Ca$^{2+}$-free apo state (Fig. 3D) and the verruculogen-bound state (Fig. 4F). At the same time, the emodepside occupies the position where incoming K$^+$ ions would be focused before entering the selectivity filter and adds another layer of the pore through which the ions have to be translocated.

several different channels. They are supposed to bind into the S6 pocket in a similar manner as verruculogen, but unlike verruculogen and emodepside, they are thought to activate the channels by suboptimal coordination of potassium in the S6 position[49]. In the absence of experimental structural information, it is not clear where the discrepancy to the more complex activity of emodepside derives from. However, we speculate that while the rigidity of the emodepside ring and the consequent imperfect K$^+$ coordination might cause the observed low ion translocation, the activation of ion translocation by NCA compounds might be due to the distribution of the coordination sites onto four NCA compound molecules which might be structurally more flexible and accommodating.

One interesting detail about the action of verruculogen and emodepside, which has also been observed previously for the

*C.elegans* homolog of Slo[44], is their competitive effect on Slo in patch-clamp assays: Verruculogen-mediated inhibition of Slo that had been saturated with emodepside is orders of magnitude lower (Supplementary Fig. 8A) than without emodepside saturation (Supplementary Fig. 5A). Conversely, the voltage-independent basal conductance mediated by emodepside (Supplementary Fig. 6A) was absent when Slo had been incubated with verruculogen first (Supplementary Fig. 8B). The reason for this is obvious from our structures as the partially overlapping binding sites of both compounds (Supplementary Fig. 8C, D) make simultaneous binding impossible. Furthermore, these results indicate that the off rates of both compounds cannot be very high since, during the course of the patch-clamp experiments, neither compound was able to substantially replace the pre-bound respective other compounds.

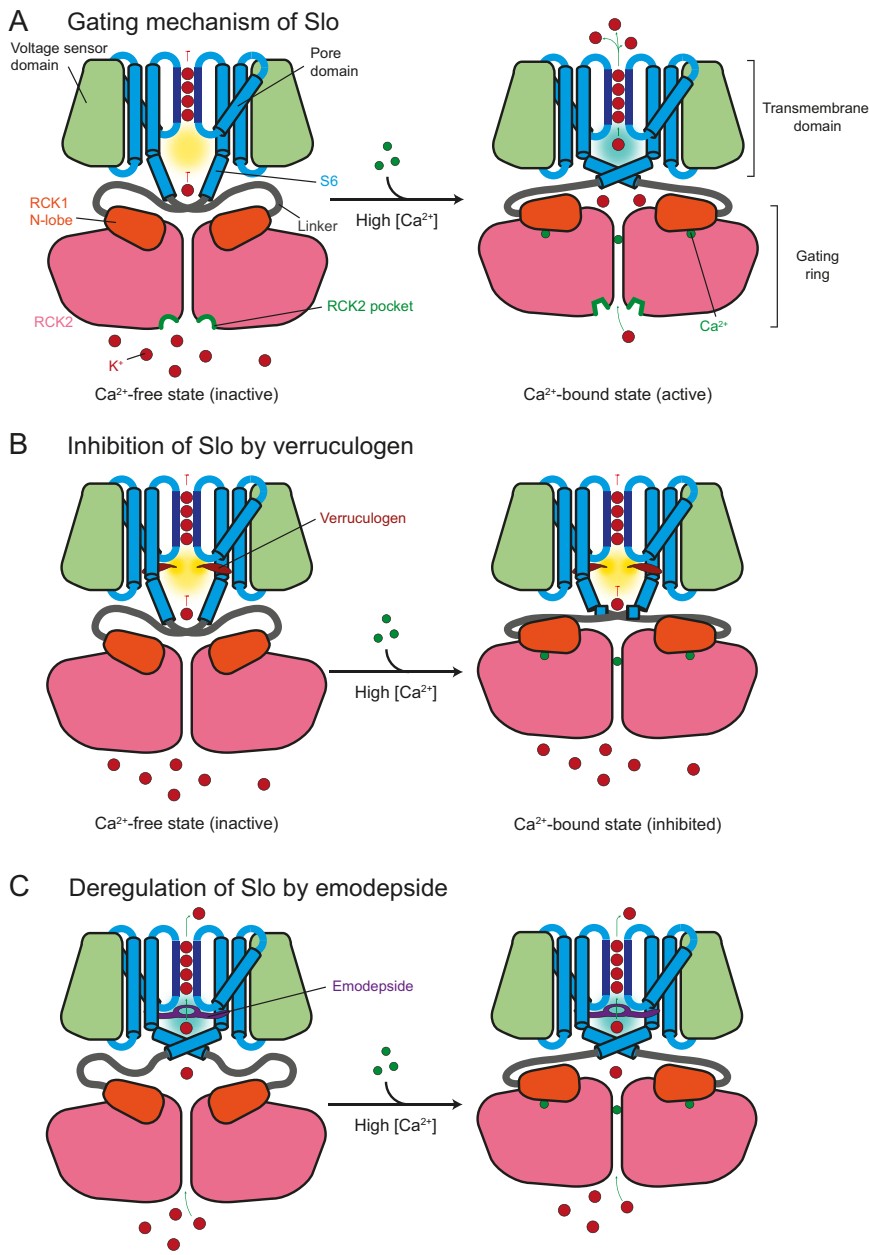

**Fig. 6 Verruculogen and emodepside modulate the activity of Slo. A** Schematic representation of Slo in the $Ca^{2+}$-free inactive (left) and $Ca^{2+}$-bound active (right) conformations. Upon $Ca^{2+}$ binding to the gating ring, the RCK1 N-lobes move outward, leading to a rearrangement of helix S6 in the TMD. This rearrangement widens the entry to the cavity and buries hydrophobic patches along its walls. Thus, $K^+$ ions can access the selectivity filter and be translocated to the extracellular space. The predicted RCK2 pocket (green) also changes its conformation upon $Ca^{2+}$-binding and offers the possibility of locking either conformation by novel small molecules. **B** Verruculogen (red) binds to the cavity pocket which is only exposed in the $Ca^{2+}$-free state and prevents helix S6 from adopting the active conformation upon $Ca^{2+}$ binding. This, together with the hydrophobic environment in the cavity which is enhanced by verruculogen precludes $K^+$ ions from reaching the selectivity filter and being translocated. **C** Emodepside (purple) binds to the intracellular entryway in the cavity and restricts access to the selectivity filter by suboptimal $K^+$ coordination. As emodepside likely stabilizes the active conformation of helix S6 also in the $Ca^{2+}$-free state, $K^+$ translocation is effectively uncoupled from regulation by $Ca^{2+}$ concentration and the transmembrane potential.

**Ligand-bound structures will facilitate the design of novel drugs acting on Slo.** In conclusion, we identified a novel potential insect-specific drug-binding pocket, termed RCK2 pocket, and provide structural and functional data that explain how Slo activity can be efficiently modulated by small molecules (Fig. 6). Verruculogen acts as a classical inhibitor that prevents K $^+$ ions from reaching the selectivity filter (Fig. 6B). In contrast, the mechanism of action of the emodepside is more complex. The

compound is positioned in a way that it could coordinate and stabilize a non-hydrated $K^+$ ion in the S6 position inside the central cavity where, in the absence of emodepside, hydrated $K^+$ are focused and positioned to be stripped off their hydration shell (Fig. 6C). Since our Cryo-EM data were acquired in the presence of $Ca^{2+}$, it is not clear how emodepside uncouples ion translocation from activation by intracellular $Ca^{2+}$ concentrations and voltage sensing. While it is tempting to speculate that emodepside

might stabilize the active conformation of the pore domain Slo in which the cavity pocket is closed even in the absence of $Ca^{2+}$, future studies are required to experimentally test this hypothesis.

This work lays the foundation for the design of novel modulators of Slo on a rational basis with characteristics that differ from the existing compounds. For example, derivatives of verruculogen might have a subtler inhibitory effect on Slo by reducing the size and hydrophobicity of the tail reaching into the central cavity. Molecules analogous to emodepside might be designed in a way that they could coordinate $K^+$ ions more efficiently to increase conductance, or less efficiently to decrease conductance, thereby modulating the effect on motor neuron function. Importantly, agrochemists will be able to use the structure and composition of the RCK2 pocket to design or test compounds that were previously not considered as insecticides.

As many cation channels possess a similar architecture including a central cavity adjacent to the selectivity filter and share a similar gating mechanism involving opening and closing of the S6 helix[14,50,51], we expect that our findings pave the road for the rational design of compounds with modulating effects also on these channels.

## Methods

**Expression and purification of *Drosophila* Slo for Cyro-EM studies.** Slowpoke, isoform J from *Drosphila* (NCBI reference sequence NP_001014658.1; https://www.ncbi.nlm.nih.gov/protein/NP_001014658.1) encoding amino acid residues 1 to 1180 carrying mutations N281D, M328I, S332C, E338D, V340I, S342T, G343R, N344A, E349T, R352J, H354K, K356R, and S974G was synthesized and codon optimized for expression in insect cells (Geneart, Thermo Fisher) and subcloned (EcoRI/BamHI) into a pOET2 expression vector (Oxford Expression Technologies). A Rho1D4 tag is fused to the C-terminus utilizing a linker stretch (GSSG). The optimized DNA sequence of the fusion construct is shown in Supplementary Table 1. The construct was expressed in *Trichoplusia ni* (*T.ni*) insect cells using the Flash Bac Ultra kit (Oxford Expression Technologies). $2 \times 10^6$ cells/ml were infected with baculovirus (2 MOI), cultivated in suspension in Insect-XPRESS medium (Lonza) at 25 °C and cells were collected 48 h after infection by centrifugation. The cell pellet was supplemented with protease inhibitors using 20 mM Hepes, 100 mM NaCl, 2 mM EDTA completed with 10 µM Leupeptin, 1 mM o-Phenanthroline, 0.1 mM phenylmethylsulfonyl fluoride (PMSF); 10 µM E-64 (N-[N-(L-3-*trans*-Carboxyoxiran-2-carbonyl)-leucyl]-agmatin, N-(*trans*-epoxy-ysuccinyl)-L-leucin-4-guanidinbutylamid) and 1 mM Pepstatin A and stored at −80 °C until further use. All protease inhibitors and chemicals were purchased at Carl Roth, Germany.

The purification of Slo was modified according to ref. [16]. All purification procedures were carried out either on ice or at 4 °C. Briefly, cells were gently disrupted by stirring in a hypotonic solution containing 10 mM Tris-HCl pH 8.0, 4 mM DTT, 2 mM EDTA supplemented with 10 µM Leupeptin, 1 mM o-Phenanthroline, 0.1 mM PMSF, 10 µM E-64 and 1 mM Pepstatin A. The cell lysate was centrifuged for 30 min at 30,000 × g and the membrane pellet was homogenized in 20 mM Tris-HCl pH 7.7, 320 mM KCl, 10 mM $CaCl_2$, 10 mM $MgCl_2$ supplemented with 10 µM Leupeptin, 1 mM o-Phenanthroline, 0.1 mM PMSF, 10 µM E-64 and 1 mM Pepstatin A. Solubilization was done with 1% (w/v) Lauryl Maltose Neopentyl Glycol (LMNG) and 0.1% (w/v) CHS (Anatrace) for 120 min and subsequently, the sample was centrifuged at 30,000 x g for 60 min. The supernatant was batch incubated with Rho1D4 agarose (Cube Biotech) o/n and extensively washed with washing buffer containing 0.003% (w/v) LMNG, 0.0003% (w/v) CHS, 20 mM Tris-HCl pH 7.7, 320 mM KCl, 10 mM $CaCl_2$, 10 mM $MgCl_2$, 10 µM Leupeptin, 1 mM o-Phenanthroline, 0.1 mM PMSF, 10 µM E-64 and 1 mM Pepstatin A. For the last incubation and washing step, insect lipids (100 µg/ml) were added to the buffer. Total insect lipids were prepared by lipid extraction of un-transfected *T.ni* cells. $2 \times 108$ cells were incubated with chloroform/methanol/water (10:5:1) (v/v/v) and lipids were extracted in a shaking water bath at 37 °C o/n. To separate insoluble cell fragments, the extraction was then separated using a separatory funnel. The chloroform phase was evaporated and the amount of total lipids was determined.

Slo was eluted from the Rho1D4 agarose by incubating 2 h with washing buffer supplemented with 1 mM Rho1D4 peptide (Cube Biotech) and 100 µg/ml insect cell lipids. Elution fractions were collected, concentrated (Amicon Ultra-15, Merck Millipore) and further purified by size exclusion chromatography on a Superose 6 increase column (GE Healthcare) in 0.003% (w/v) LMNG, 0.0003% (w/v) CHS, 20 mM Tris-HCl pH 7.7, 320 mM KCl, 10 mM $CaCl_2$, 10 mM $MgCl_2$, 10 µM Leupeptin, 1 mM o-Phenanthroline, 0.1 mM PMSF, 10 µM E-64, and 1 mM Pepstatin A. The size exclusion buffer did not contain insect lipids. The peak fraction corresponding to the tetrameric Slo protein was concentrated to 3 mg/ml and used for cryo-EM grid preparation.

**Grid preparation.** Grids were prepared using a Vitrobot Mark IV (Thermo Fisher Scientific) at 13 °C and 100% humidity. 4 µl of a 2–3 mg/ml Slo solution were applied to glow-discharged UltrAuFoil R2/2 200 grids (Quantifoil) and excess liquid removed by blotting (3.5 s at blot force −3) before vitrification in liquid ethane.

The $Ca^{2+}$-free Slo apo sample was prepared by incubating 3 mg/ml Slo (purified as described above in a buffer containing Calcium) with 50 mM EDTA for 15 min on ice prior to grid preparation.

Verruculogen (Cayman Chemical Cooperation) was dissolved in DMSO at 30 mM. 2 mg/ml Slo (equaling 15 µM Slo monomer) was incubated with 30 µM verruculogen for 3 h on ice; the final DMSO concentration was 1% (v/v).

Emodepside (synthesized at Bayer Animal Health, now Elanco) was dissolved in DMSO at 30 mM. 2 mg/ml Slo (equaling 3.75 µM Slo tetramer) was incubated with 30 µM emodepside for 3 h on ice; final DMSO concentration was 5% (v/v).

To prepare the $Ca^{2+}$-bound apo sample, 2 mg/ml Slo was incubated with 20 µM Aflatrem for 3 h on ice; final DMSO concentration was 1% (v/v). Aflatrem was purchased from WuXi AppTech and dissolved in DMSO at 30 mM. However, since no additional density of the compound was visible in the reconstructions, and the reconstruction and molecular model were virtually identical to a 3.0 Å reconstruction we had obtained earlier in the absence of Aflatrem, we decided to treat the higher quality reconstruction as apo and the sample in presence of Aflatrem as $Ca^{2+}$-bound apo sample.

**Cryo-EM data acquisition.** Cryo-EM datasets of the different Slo samples in the absence and presence of ligands were acquired on a Titan Krios electron microscope (Thermo Fisher Scientific) equipped with a field emission gun. A total of 5122 ($Ca^{2+}$-bound apo), 5467 ($Ca^{2+}$-free apo), 7007 (verruculogen) and 11,135 (emodepside) movies were recorded on a K3 camera (Gatan) operated in super-resolution mode at a nominal magnification of 130,000, resulting in a super-resolution pixel size of 0.35 Å. A Bioquantum post-column energy filter (Gatan) was used for zero-loss filtration with an energy width of 20 eV. Total electron exposure of 74–80 e−/ Å$^2$ was distributed over 60 frames. Data were collected using the automated data collection software EPU (Thermo Fisher Scientific), with five acquisitions per hole and a set defocus range of −0.9 to −2.1 µm.

Details of data acquisition parameters can be found in Supplementary Table 2.

**Cryo-EM data processing.** For all datasets, data processing procedures followed highly similar strategies. In each case, on-the-fly data preprocessing and quality control was performed within TranSPHIRE[52]. RELION[53] was used for correcting beam-induced motion and dose-weighting, and CTF parameters were estimated using CTFFIND4[54] operated in movie mode. Particles were picked by SPHIRE-crYOLO using a generalized neural network trained on multiple datasets of unrelated samples[55]. Data acquisition and processing are summarized in Supplementary Table 2.

For the $Ca^{2+}$-bound apo structure, 906,619 2-fold binned, crYOLO-picked particles from 5097 micrographs (the remaining 25 micrographs did not contain particles) were extracted using SPHIRE[56] with a box size of 180 × 180 pixels and used for 2D classification in ISAC[57] using a particle radius of 75 pixels and classes with maximally 2000 members. An initial model was calculated by RVIPER from 117 high-quality 2D class averages with imposed C4 symmetry. 3D reconstruction in MERIDIEN was performed, also imposing C4 symmetry, using 226,076 particles assigned to well-defined 2D classes, resulting in a 2.8 Å 3D reconstruction. In parallel, micrographs were inspected using the CTF assessment tool in SPHIRE, and micrographs with too high (>2.8 µm) or too low (<0.5 µm) defocus or too high CTF anisotropy (>0.2 µm) were removed. A total of 225,754 centered particles were re-extracted from the remaining 4830 micrographs without binning using a box size of 360 × 360 pixels. These were used for masked 3D refinement in MERIDIEN, yielding a 2.6 Å reconstruction. This was followed by several iterative rounds of Bayesian polishing and CTF refinement in RELION, improving the resolution of the reconstruction to 2.4 Å. Particles were then re-extracted using a box size of 448 × 448 pixels and converted into a SPHIRE stack. A final 3D refinement in MERIDIEN using this final set of polished particles and optimized CTF parameters yielded a 2.38 Å reconstruction. The final reconstruction for real-space refinement was obtained by combining both half-volumes and applying a sharpening B-factor of 33.8 Å$^2$ in SPHIRE, followed by a round of PHENIX.Autosharpen. A map extending to 2.34 Å showing slightly more high-resolution features was obtained by three rounds of density modification using PHENIX.Resolve[58]. Details about processing of the $Ca^{2+}$-bound apo Slo Cryo-EM data and analysis of the final reconstruction can be found in Supplementary Fig. 9.

For the $Ca^{2+}$-free apo structure, 920,897 2-fold binned, crYOLO-picked particles from 5413 particle-containing micrographs were extracted using SPHIRE[56] with a box size of 180 × 180 pixels and used for 2D classification in ISAC[57] using a particle radius of 75 pixels and a class size limit of 2000. RVIPER was used to generate an initial 3D model from 58 high-quality 2D class averages with imposed C4 symmetry. 3D reconstruction in MERIDIEN was performed, using 96,557 particles assigned to well-defined 2D classes, resulting in a 3.1 Å 3D reconstruction with imposed C4 symmetry. This subset of well-centered high-quality particles was used for training a new model in crYOLO which was then used for the second round of particle picking. A new stack of 363,787 particles was extracted using a box size of 360 × 360 pixels and subsequently subjected to one

round of stringent cleanup in ISAC with a particle radius of 140 px and a class size of 1000, resulting in a substack of 90,833 particles assigned to 2D classes showing high-resolution features. These were used for masked 3D refinement in MERIDIEN, yielding a 3.0 Å reconstruction. This was followed by several iterative rounds of Bayesian polishing and CTF refinement in RELION, improving the resolution of the reconstruction to 2.6 Å. Particles were then re-extracted using a box size of 448 × 448 pixels and converted into a SPHIRE stack. A final 3D refinement in MERIDIEN was performed using an optimized, looser mask. This run yielded a slightly lower nominal resolution but included several weak-density features including parts of helix S0 in the VSD and the linker between the pore domain and RCK1 which were omitted by the tight mask in earlier reconstructions. Two final reconstructions for real-space refinement and model building were obtained, either at 2.68 Å by combining both half-volumes and applying a sharpening B-factor of 26.1 Å² in SPHIRE, followed by a round of PHENIX.Autosharpen, or at 2.62 Å by three rounds of density modification using PHENIX.Resolve[58]. Details about processing of the $Ca^{2+}$-free apo Slo Cryo-EM data and analysis of the final reconstruction can be found in Supplementary Fig. 10.

A total of 1,131,410 particles were picked with crYOLO on 6792 particle-containing micrographs of the verruculogen dataset and extracted with a box size after 2-fold binning of 180 × 180 pixels. 2D classification in ISAC[57] was performed with a particle radius of 75 pixels and 2000 particles per class. 93 high-quality 2D class averages were used for initial model calculation in RVIPER with imposed C4 symmetry. 3D reconstruction in MERIDIEN was performed with 153,173 particles assigned to well-defined 2D classes and C4 symmetry, and resulted in a 3.4 Å 3D reconstruction. New particle coordinate files were created using the projection parameters of this MERIDIEN run and were used for training a new model in crYOLO. Using this model, a new stack of 404,306 particles was extracted from 6522 micrographs using a box size of 360 × 360 pixels, and subsequently subjected to two consecutive rounds of the cleanup in ISAC with a particle radius of 150 px and class sizes of 1000 and 200, respectively, leaving a substack of 127,912 particles assigned to 2D classes showing high-resolution features. This substack was subjected to masked 3D refinement in MERIDIEN, yielding a 3.0 Å reconstruction. This was followed by several iterative rounds of Bayesian polishing and CTF refinement in RELION, improving the resolution of the reconstruction to 2.7 Å. A 3D classification run resulted in one of the three classes displaying the best-ordered verruculogen density. 66,077 particles associated with this class were then re-extracted using a box size of 448 × 448 pixels and converted into a SPHIRE stack and a final 3D refinement run was performed in MERIDIEN. The final reconstruction was obtained by combining both half-volumes, filtering to 2.73 Å and applying a sharpening B-factor of 23.8 Å² in SPHIRE, followed by a round of PHENIX.Autosharpen. Three rounds of density modification using PHENIX.Resolve[58] resulted in a second map at 2.64 Å which was used to help manual model building in COOT. Details about processing of the verruculogen-bound Slo Cryo-EM data and analysis of the final reconstruction can be found in Supplementary Fig. 11.

A total of 1,861,584 crYOLO-picked particles were extracted from 11,133 micrographs of the emodepside dataset with a box size without binning and a box size of 360 × 360 pixels. This particle stack was cleaned by 2D classification in ISAC with 150 px particle radius and 1000 particles per class. The resulting substack included 261,597 particles associated with 278 high-quality classes. 3D refinement in MERIDIEN was initially performed without imposing symmetry using this subset and an initial reference model created with RVIPER, and resulted in a 2.9 Å resolution reconstruction. The particle stack was then transferred into RELION and several rounds of Bayesian polishing and CTF refinement were performed. Inspection of the resulting 2.6 Å reconstruction revealed an additional density attributed to emodepside that indicated 2-fold symmetry. Therefore, subsequent reconstructions were performed with imposed C2 symmetry and 448 voxel box size. An additional round of Bayesian polishing improved the resolution of the reconstruction to 2.5 Å. As the global 3D refinement in MERIDIEN and RELION was not able to completely resolve the symmetry mismatch between C4 symmetry in large parts of Slo and the local reduction to C2 symmetry by and around emodepside, we created a focus mask including only the density corresponding to the pore domains. Using this mask, a local refinement in RELION was performed in C2 symmetry, but relaxing the symmetry to C4, yielding a reconstruction with a nominal resolution of 2.5 Å. The particles with correctly assigned projection parameters were used for a final local refinement in MERIDIEN and postprocessing using a sharpening B-factor of 36.3 Å² followed by PHENIX Autosharpening, resulting in a reconstruction of slightly lower nominal resolution (2.59 Å), but with more homogeneous local resolution throughout the whole Slo particle. Three rounds of density modification using PHENIX.Resolve[58] resulted in a second map at 2.50 Å which was used to help manual model building in COOT. Details about processing of the emodepside-bound Slo Cryo-EM data and analysis of the final reconstruction can be found in Supplementary Fig. 12.

**Model building and visualization**. An initial homology model of *Drosophila* Slo was generated with SWISS-MODEL[59] using *Aplysia* Slo1 (PDB 5TJ6; https://doi.org/10.2210/pdb5TJ6/pdb)[16] as a model. This homology model was docked into the 2.34 Å density-modified Coulomb potential map of Calcium-bound apo Slo by rigid-body fitting. Then, this initial model was iteratively improved by cycles of manual building in COOT[60] and real-space refinement in PHENIX[58]. Thereby,

the auto sharpened map was used in PHENIX and the map derived from density modification was used in COOT. During the first real-space refinement run, one round of simulated annealing was performed with secondary structure restraints in place, whereas in later runs, only smaller scale morphing was performed without using secondary structure restraints. NCS constraints were used throughout all real-space refinement runs. The final model comprises residues L44-G1145, but does not contain the N- and C-terminal tails (residues M1-C43 and L1146-S1180 plus the GSSG linker and the Rho1D4 tag, respectively) as well the linker between helices S0 and S0' of the VSD (residues C78-G105), one short loop in RCK1 (residues G587-D593), the long linker between RCK1 and RCK2 (residues C631-E777) and two loops in RCK2 (residues D928-S958 and S1110-D1113, respectively) which were all disordered. Four spherical densities along the symmetry axis inside the central pore were modeled as $K^+$ ions with partial occupancy (0.5 and 0.25, respectively). Two $Ca^{2+}$ ions per Slo monomer were modeled in the previously characterized binding sites in the gating ring[16,17,23,24]. In addition, another density that was not identified in the previous homologous structures was found close to the RCK1 calcium-binding site and was modeled as $Mg^{2+}$ ion. Conversely, the previously characterized magnesium binding site between the gating ring and the VSD[16,17] was not present in our structure, probably because the asparagine in the linker between helices S2 and S3 (N237 in the human protein) that makes the closest contact from the VSD side is serine in *Drosophila* (S187; Supplementary Fig. 1). Several elongated densities surrounding the TMD, some of them overlapping with lipid-binding sites in humans and *Aplysia* Slo1[16,17], were assigned as ordered lipid molecules and modeled as phosphatidylcholine as the most abundant lipid in insect cells[61]. In addition, one ordered cholesterol per Slo was modeled.

The apo model was used for model building of all other conformations and adjusted to the respective maps by iterative rounds of manual building in COOT and real-space refinement in PHENIX. Restraints for verruculogen refinement were generated using JLigand[62] in CCP4[63] using a published crystal structure[64] as a starting model. Refinement restraints for emodepside were generated, along with an initial de novo model, by using CORINA (Altamira LLC).

In the $Ca^{2+}$-free apo reconstruction, the VSD is only badly resolved compared to the rest of the protein, which indicates increased flexibility of this domain. Especially for the C-terminal part of helix S0 (residues I57-C77) and the complete helix S0' (residues T106-G116), only very faint residual density was visible after density modification. Nevertheless, we have modeled the respective pieces of the structure into this residual density for the purpose of more consistent structure interpretation and illustration, guided by the model of the calcium-bound apo structure as well as the calcium-free human Slo structure (PDB 6V3G; https://doi.org/10.2210/pdb6V3G/pdb)[17].

All structural figures and supplementary movies were prepared using UCSF ChimeraX[65]. Surface hydrophobicity as shown in Figs. 3C, D, 4F and 5F were calculated using the MLPP[66] implementation in ChimeraX. The pore diameter plots (Figs. 4E and 5E, Supplementary Fig. 2H) were created using HOLE[67]. Statistics to assess model quality can be found in Supplementary Table 2.

**Electrophysiology**. Verruculogen (Cayman Chemical Cooperation) and emodepside (synthesized at Bayer Animal Health, now Elanco) were dissolved at 10 mM in DMSO and stored at −20 °C. Prior to measurements, they were freshly diluted to the required concentrations in external solution with a final concentration of 1% DMSO and 0.03% Pluronic (Sigma Aldrich, Poloxamer 188 solution); for automated patch clamp measurements, the solution contained in addition 0.05% BSA (Sigma Aldrich). All other chemical reagents were purchased from Sigma Aldrich if not stated otherwise. For conventional patch-clamp recordings, the external ringer bath-solution contained 150 mM NaCl, 4 mM KCl, 2 mM $MgCl_2$, 2 mM $CaCl_2$, and 10 mM HEPES-NaOH pH 7.4, and the internal pipette solution contained 150 mM KCl, 3 mM or 7.5 mM $CaCl_2$, respectively, 10 mM EGTA and 10 mM HEPES-KOH pH 7.2. For automated patch clamp the external solution contained 140 mM NaCl, 4 mM KCl, 1 mM $MgCl_2$, 2 mM $CaCl_2$, 10 mM HEPES-NaOH pH 7.4 and 5 mM glucose. Sealing was promoted by Seal-Enhancer containing 80 mM NaCl, 35 mM $CaCl_2$, 10 mM $MgCl_2$, 3 mM KCl, 10 mM HEPES-NaOH pH 7.4. Intracellular solution for patching contained 60 mM KF, 75 mM KCl, 1 mM $MgCl_2$, 2 mM $Na_2$-ATP, 10 mM HEPES-KOH pH 7.2 and 2 mM EGTA. During measurements, internal solution was exchanged to low $Ca^{2+}$ buffer containing 85 mM KCl, 40 mM KF, 3 mM $CaCl_2$, 1 mM $MgCl_2$, 2 mM $Na_2$-ATP, 10 mM HEPES-KOH pH 7.2 and 10 mM EGTA or high $Ca^{2+}$ buffer containing 83 mM KCl, 50 mM KF, 2 mM $CaCl_2$, 1 mM $MgCl_2$, 2 mM $Na_2$-ATP, 10 mM HEPES-KOH and 2 mM EGTA.

Automated patch-clamp recordings were performed to obtain dose-response curves, using a SyncroPatch384 device (Nanion Technologies). For measurements on CHO cells stably expressing *Drosophila* Slo (GenBank: AAA28651.1)[38], single-hole high resistance chips (10 MΩ) were used. Pulse generation and data collection were performed with PatchController384, evaluated via DataController384 software (Nanion Technologies) and plotted using GraphPad Prism. Prior to patch-clamp experiments, cells that were grown in 175 cm² cell culture flasks were detached with accutase® solution and kept in the SynchroPatch device at 10 °C and 60 rpm shaking. Cells were transferred to and caught within the measurement chambers and sealed by adding Seal-Enhancer solution. Then, measurement solutions were applied. Whole-cell recordings were conducted by subsequent exchange of internal solution, first compound application (e.g., verruculogen), second compound application (e.g., emodepside) and finally 3 mM $BaCl_2$ application as control. To obtain dose-response

curves a voltage ramp from −160 mV to 120 mV within 170 ms was applied and repeated every 10 s, the holding potential was −100 mV. For analysis, the currents at the ramp potential of +100 mV were used.

All representative traces shown were obtained by manual patch-clamp recordings, conducted in the whole-cell voltage-clamp configuration as described elsewhere[68]. Patch pipettes were pulled from borosilicate glass capillaries (GB150-8P 0.86 × 1.50 × 80 mm) to a tip resistance of 2–4 MΩ. CHO cells stably expressing *Drosophila* Slo (GenBank: AAA28651.1) were cultured on coverslips coated with Poly-D-Lysin. Compounds were applied using the U-tube reversed flow technique[69]. The voltage protocol consisted either of a ramp from −160 mV to 100 mV within 170 ms, or pulses from −120 mV to 60 mV in steps of 10 mV, the holding potential was clamped to −100 mV. Data were collected at 10 kHz and 3 kHz low-Bessel filtered using an EPC-10 amplifier (HEKA electronic) and PatchMaster software (HEKA electronic), and plotted using IgorPro (WaveMetrics).

**Reporting Summary**. Further information on research design is available in the Nature Research Reporting Summary linked to this article.

## Data availability

The data that support this study are available from the corresponding author upon reasonable request. The atomic coordinates and cryo-EM maps for Slo alone and in complex with verruculogen and emodepside are available at the Protein Data Bank (PDB)/Electron Microscopy Data Bank (EMDB) databases. The accession numbers are 7PXE and EMD-13700 for Ca$^{2+}$-bound apo, 7PXF and EMD-13701 for Ca$^{2+}$-free apo, 7PXG and EMD-13702 for verruculogen, and 7PXH and EMD-13703 for emodepside, respectively. Source data are provided with this paper.

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

## Acknowledgements
We thank Daniel Prumbaum for support with cryo-EM data acquisition and Klaus Raming for providing the stable CHO cell line expressing *Drosophila melanogaster* Slo. This work was funded by the Max Planck Society (to S.R.).

## Author contributions
J.F. and S.R. conceived and managed the project; J.K and B.M. expressed and purified the protein; T.R. prepared specimens, and recorded and processed cryo-EM data; T.R. and O.G. analyzed the structures; O.H. recorded cryo-EM data; A.B and U.E.-K. generated and analyzed electrophysiological data; T.R., A.B. and U.E.-K. prepared figures; and T.R. and S.R. wrote the manuscript with input from all authors.

## Funding

## Competing interests
J.F. and U.E.-K. are shareholders of Bayer A.G., J.K., and B.M. are shareholders of Cube Biotech GmbH. The other authors declare no competing interests.
