## [Peer Review File · Nature Communications]

Small molecule modulation of the Drosophila Slo channel elucidated by cryo-EMREVIEWER COMMENTS

Reviewer #1 (Remarks to the Author):

In this report by Raisch and colleagues, the authors present structures of the *Drosophila* Slo K⁺ channel in ligand-free open and closed states as well as in the presence of verruculogen and emodepside. This work builds upon previous structural and functional studies of Slo channels to determine the mechanisms of activation by emodepside and inhibition by verruculogen. These molecules both bind in the pore but have divergent effects on conformation. Emodepside locks the channel into an active state, while verruculogen prevents opening of the channel gate. Structures in the absence of ligand also provide potential targets for novel small molecule regulators that could be specifically targeted against insects. Together, these studies represent an important advancement in the understanding of Slo channel regulation by small molecules and this work is suitable for publication in *Nature Communications*.

Comments

1. As the authors point out, one of the benefits of achieving high-resolution with their ligand-free structures is that these structures can serve as the basis for rational design of novel small molecules. The authors identify several potential binding sites in the RCK domains, including a site they term the RCK2 pocket. It would be helpful for the readers to visualize how well conserved are the residues that line this pocket in insects compared to mammalian Slo channels.
2. It would be similarly helpful to visualize the conservation of the emodepside and verruculogen binding sites, especially as the structures of human Slo, *Drosophila* Slo and *Aplysia* Slo all closely resemble one another and are gated by the same mechanisms. This would guide future design studies by resolving which pockets would be universal and which ones might be suitable for more selective molecules.

Reviewer #2 (Remarks to the Author):

Raisch et al. present new structures of Slo (BK) Ca²⁺ gated potassium channels from *Drosophila*. Two of these structures reiterate the previously known gating mechanism of these channels from other organisms - *Aplysia* and humans. Two remaining structures show binding of two active small molecules - verruculogen and emodepside. Both of them bind in the so called central cavity of BK channels, below the selectivity filter. Verruculogen seems to bind to the closed (Ca²⁺-free) state of the channel, and locks it in this conformation. Emodepside however binds to the open (Ca²⁺-bound) state of the channel, locks it in this conformation, effectively uncoupling it from Ca²⁺ and voltage. The ring of emodepside creates then an additional binding site of potassium below the selectivity filter, however we arguably worse coordination of a K⁺ ion, to which the authors attribute lower currents of BK channels in the presence of emodepside.

The study is well-designed and the new structures provide important insights on drug binding to BK channels, and provide hypotheses how these drugs can actually work on said channels. Therefore, in my opinion, these structures should definitely be published. I was asked to comment on MD simulations, and unfortunately their quality do not follow the quality of the structural work in the manuscript. My detailed list of criticisms and possible fixes can be found below; however, given a minor contribution of MD simulations to the whole manuscript, I'd suggest to remove them completely - in my opinion the manuscript will be still strong enough to justify its publication. Alternatively the authors could contact one of the groups that specialize in MD simulations of ion channels.

If the authors however choose to keep the MD simulations, the following points would need to be addressed in the revised version:

1. There is no numerical analysis of MD trajectories in the manuscript. The results from MD are based

on visual inspection only, saying that there is fewer molecules in the cavity of the channel, when verruculogen is bound. Note that a proper analysis would require not only counting the number of water molecules in the cavity, but also providing properly statistically treated estimates. Given only single trajectories that are relatively short (tens of nanoseconds) that might be difficult with the current set of trajectories.

2. The authors state in the text that verruculogen locks the channel in a conformation similar to the closed conformation, but then in MD compare the outcome of verruculogen-bound system to the open (Ca²⁺-bound) conformation. It seems to me like comparing apples with oranges. If verruculogen really locks the channel in the closed conformation, that would explain on its own its inhibitory effect. Of interest, these BK channels have been actually postulated to gate through hydrophobic gating in the central cavity in the closed (Ca²⁺-free) conformation, so to see any water molecules in the cavity is actually surprising (see Jia et al., Nat Comm 2018).

3. As mentioned, sampling times are quite short, so all these observations might suffer from insufficient sampling. At least few hundreds ns long trajectories, in several replicates, would be required to obtain physically and statistically meaningful insights. Moreover, hydration/dehydration of small hydrophobic cavities at the protein/water/membrane interface poses a big challenge for current, not so accurate force fields (see papers from Mark Sansom lab). Therefore, the usage at least two force fields would be welcome.

4. The effects of cavity hydration/dehydration can be further influenced by the fact of using a truncated version of the channel. Such a model should be at least validated by comparison of the overall conformation (e.g. RMSD) to the experimental structure over the course of MD simulations.

5. It is not clear to me why position restraints have been used on the protein ends. What do the authors mean by 'drifting'? Is the tetramer unstable? If its simply drifting away from the box center, the protein can be recentered in the post-processing step.

The details of MD simulations are missing - what was the lipid and protein force field? What parameters and algorithms have been used?

Other comments:

1. The authors seem to use quite high concentration of verruculogen in the experiment, and end up with four molecules bound to the channel. Is it something to be expected to occur physiologically, or is it possible that only 1, 2 or 3 molecules might be bound and yet show their inhibitory effect? Did the authors try to get the Hill coefficient of verruculogen binding?

2. Some description of the channel and its presentation is somewhat confusing and do not follow a typical presentation of potassium channels. The "pre chamber" is usually called a (central) cavity. The channels are usually presented with the extracellular side being on top. The ion binding sites in the selectivity filter have their names (S1-S4) together with additional binding sites - S0, Scav (from cavity) - that might not be present in the current structures due to low resolution, but might be nevertheless important for ion permeation in BK channels. That's particularly important for the discussion of emodepside, as it seems it might overlap with the Scav binding site.

3. The hypothesis that emodepside reduces the current through BK channels by creating a sub-optimal potassium binding site below the SF is interesting, especially given that similar mechanism have been proposed for BL (NCA) compounds to actually enhance the current through BK channels (see Schewe et al., Science 2019). It would be of interest if the authors could discuss similarities and differences between these two class of compounds.

Reviewer #3 (Remarks to the Author):

This article by Raisch et al. reported four cryoEM structures of the *Drosophila* Slo channel in various functional states. These structures revealed potential insect-specific binding pockets and the binding modes of two small molecules, verruculogen and emodepside. Based on these observations, the authors proposed mechanisms for verruculogen and emodepside inhibition. To further support their hypotheses, the authors performed MD simulations to investigate the effect of verruculogen on ion permeation. The cryoEM analyses seem solid and the resolutions of reported structures are sufficiently high for the structural interpretation. The proposed models of verruculogen and emodepside are interesting. This work could be a useful addition to the ion channel field and I would recommend its publication if the following points are addressed or discussed.

Major

1. The effects of emodepside on Ca²⁺ and voltage sensing are very interesting. The abstract made me think that the authors have found the underlying mechanisms and I was disappointed to see that the emodepside-bound structure failed to explain how that happens. Determining structures to explain voltage sensing is difficult and beyond the scope of this study. On the other hand, determining an emodepside-bound structure in the absence of Ca can potentially provide more insights into Ca²⁺ sensing part. But I also understand that is a lot of work. If getting another structure is challenging, the author should at least explicitly discuss the limitation of this study regarding emodepside's modulation on Ca²⁺ and voltage sensing.
2. Since the proposed mechanisms of verruculogen and emodepside involve ions in the filter, please show the cryoEM density of K⁺ ion and H₂O (if visible, like those in Supplementary Fig .6G) in the filter and prechamber, with surrounding amino acids contoured at the same level if possible.

Minor

1. In the abstract, emodepside is described first and then verruculogen ("Emodepside locks the transmembrane domain.... and voltage sensing. Verruculogen binding inhibits...."), but in the main text, the story of verruculogen is described first. Is there a reason to reverse this order in the abstract?
2. Abstract. "Emodepside locks the transmembrane domain in an active conformation...uncoupling ion gating from Ca²⁺ and voltage sensing". Since the uncoupling effect has already been reported before (Crisford et al., 2015) and the fact that from the current study, it is difficult to understand how the drugs uncouple ion gating from Ca²⁺ and voltage sensing, I suggest revise these sentences so the reader wouldn't have wrong expectations on what questions are answered by this study.
3. The authors choose to present the channel upside down (intracellular domain on the top). This is in contrast to what most people in the field do and I do not see any benefit to present Slo channel in this way. But I will let the authors decide whether they want to follow the convention.
4. Page 6. "...and hand them down to a position between emodepside and the K⁺ in position 1 of the filter.". The position the authors referred to here should be position 4, not position 1. Position 1 is the one near the extracellular side (Zhou et al., 2001).
5. Does emodepside change the ion selectivity? From Supplementary Fig .6B it is hard to see where the reversal potential of the control group is. Although the fact that emodepside does not affect the filter structure suggests the drug does not change ion selectivity, but it would be interesting to know.

We thank the reviewers for their positive and constructive feedback, which aided us to further improve the manuscript. Major modifications of the manuscript are highlighted in yellow. Below we include our detailed response to each point raised.

Reviewer #1 (Remarks to the Author):

In this report by Raisch and colleagues, the authors present structures of the Drosophila Slo K⁺ channel in ligand-free open and closed states as well as in the presence of verruculogen and emodepside. This work builds upon previous structural and functional studies of Slo channels to determine the mechanisms of activation by emodepside and inhibition by verruculogen. These molecules both bind in the pore but have divergent effects on conformation. Emodepside locks the channel into an active state, while verruculogen prevents opening of the channel gate. Structures in the absence of ligand also provide potential targets for novel small molecule regulators that could be specifically targeted against insects. Together, these studies represent an important advancement in the understanding of Slo channel regulation by small molecules and this work is suitable for publication in Nature Communications.

Comments

1. As the authors point out, one of the benefits of achieving high-resolution with their ligand-free structures is that these structures can serve as the basis for rational design of novel small molecules. The authors identify several potential binding sites in the RCK domains, including a site they term the RCK2 pocket. It would be helpful for the readers to visualize how well conserved are the residues that line this pocket in insects compared to mammalian Slo channels.

We thank the reviewer for raising this issue, revealing that our visualization of conservation in the alignment in Supplementary Fig. 1 and close-up figures of the pockets (Fig. 2E,F, and Supplementary Fig. 3C-H) was not sufficient or well enough explained. We have now prepared an additional Supplementary Figure 4 in which we have aligned the sequences of the parts of Slo that form the predicted pockets. These alignments include various pest and beneficial insect and vertebrate species potentially exposed to agrochemical substances, and parasitic worms, and we have further analyzed these alignments for residues that differ between those species and could potentially allow for the development of specific insecticides or anthelmintics.

2. It would be similarly helpful to visualize the conservation of the emodepside and verruculogen binding sites, especially as the structures of human Slo, Drosophila Slo and Aplysia Slo all closely resemble one another and are gated by the same mechanisms. This would guide future design studies by resolving which pockets would be universal and which ones might be suitable for more selective molecules.

Please see answer to point 1; in the alignment of the S6 pocket in Supplementary Fig. 4, we have also marked the residues contacting verruculogen and emodepside, respectively.

Reviewer #2 (Remarks to the Author):

Raisch et al. present new structures of Slo (BK) Ca²⁺ gated potassium channels from Drosophila. Two of these structures reiterate the previously known gating mechanism of these channels from other organisms - Aplysia and humans. Two remaining structures show binding of two active small molecules - verruculogen and emodepside. Both of them bind in the so called central cavity of BK channels, below the selectivity filter. Verruculogen seems to bind to the closed (Ca²⁺-free) state of the channel, and locks it in this conformation. Emodepside however binds to the open (Ca²⁺-bound) state of the channel, locks it in this conformation, effectively uncoupling it from Ca²⁺ and voltage. The ring of emodepside creates then an additional binding site of potassium below the selectivity filter, however we arguably worse coordination of a K⁺ ion, to which the authors attribute lower currents of BK channels in the presence of emodepside.

The study is well-designed and the new structures provide important insights on drug binding to BK channels, and provide hypotheses how these drugs can actually work on said channels. Therefore, in my opinion, these structures should definitely be published. I was asked to comment on MD simulations, and unfortunately their quality do not follow the quality of the structural work in the manuscript. My detailed list of criticisms and possible fixes can be found below; however, given a minor contribution of MD simulations to the whole manuscript, I'd suggest to remove them completely - in my opinion the manuscript will be still strong enough to justify its publication. Alternatively the authors could contact one of the groups that specialize in MD simulations of ion channels.

We thank the reviewer for the detailed and well-justified criticism of our MD simulations. As pointed out, the MD simulations were supposed to be a minor supplementary addition to help explain the efficient inhibition of ion translocation by verruculogen. Therefore, we had decided to perform the calculations using a system reduced to the minimal core of the protein and simulating only rather short time spans. Since addressing all questions the reviewer raised and following all of their suggestions would go way beyond the scope of this study, we have decided to follow the recommendation of the reviewer and have removed the MD simulations from the manuscript.

If the authors however choose to keep the MD simulations, the following points would need to be addressed in the revised version:

- 1. There is no numerical analysis of MD trajectories in the manuscript. The results from MD are based on visual inspection only, saying that there is fewer molecules in the cavity of the channel, when verruculogen is bound. Note that a proper analysis would require not only counting the number of water molecules in the cavity, but also providing properly statistically treated estimates. Given only single trajectories that are relatively short (tens of nanoseconds) that might be difficult with the current set of trajectories.*

2. The authors state in the text that verruculogen locks the channel in a conformation similar to the closed conformation, but then in MD compare the outcome of verruculogen-bound system to the open (Ca^{2+} -bound) conformation. It seems to me like comparing apples with oranges. If verruculogen really locks the channel in the closed conformation, that would explain on its own its inhibitory effect. Of interest, these BK channels have been actually postulated to gate through hydrophobic gating in the central cavity in the closed (Ca^{2+} -free) conformation, so to see any water molecules in the cavity is actually surprising (see Jia et al., Nat Comm 2018).

3. As mentioned, sampling times are quite short, so all these observations might suffer from insufficient sampling. At least few hundreds ns long trajectories, in several replicates, would be required to obtain physically and statistically meaningful insights. Moreover, hydration/dehydration of small hydrophobic cavities at the protein/water/membrane interface poses a big challenge for current, not so accurate force fields (see papers from Mark Sansom lab). Therefore, the usage of at least two force fields would be welcome.

4. The effects of cavity hydration/dehydration can be further influenced by the fact of using a truncated version of the channel. Such a model should be at least validated by comparison of the overall conformation (e.g. RMSD) to the experimental structure over the course of MD simulations.

5. It is not clear to me why position restraints have been used on the protein ends. What do the authors mean by 'drifting'? Is the tetramer unstable? If it's simply drifting away from the box center, the protein can be recentered in the post-processing step.

The details of MD simulations are missing - what was the lipid and protein force field? What parameters and algorithms have been used?

Other comments:

1. The authors seem to use quite high concentration of verruculogen in the experiment, and end up with four molecules bound to the channel. Is it something to be expected to occur physiologically, or is it possible that only 1, 2 or 3 molecules might be bound and yet show their inhibitory effect? Did the authors try to get the Hill coefficient of verruculogen binding?

As noted by the reviewer, the concentration of verruculogen was very high, i.e. several orders of magnitude above the IC_{50} concentration determined in Crisford et al., 2015 and most likely higher than concentrations reached in any physiological context. This concentration was chosen with the purpose of saturating all binding sites in the complex to reduce heterogeneity in the EM analysis. From the analysis of the dose-response curves of our electrophysiology experiments, we have determined a Hill coefficient close to 1, indicating that verruculogen binding to the four binding sites and inhibition of potassium

translocation is likely to be non-cooperative. Thus, with the limitation that these are indirect observations and might not completely reflect the actual verruculogen binding events, we would assume that only 1, 2 or 3 molecules of verruculogen could be bound and would already show a partial inhibitory effect.

2. Some description of the channel and its presentation is somewhat confusing and do not follow a typical presentation of potassium channels. The “pre chamber” is usually called a (central) cavity. The channels are usually presented with the extracellular side being on top. The ion binding sites in the selectivity filter have their names (S1-S4) together with additional binding sites - S0, Scav (from cavity) - that might not be present in the current structures due to low resolution, but might be nevertheless important for ion permeation in BK channels. That's particularly important for the discussion of emodepside, as it seems it might overlap with the Scav binding site.

We thank the reviewer for pointing this out. We have implemented the suggested changes in the manuscript and the figures, i.e. we have turned the channels in all figures by 180 degrees and have renamed the ‘pre-chamber’ to ‘central cavity’.

As the reviewer mentions, emodepside binds close to the ‘Scav’ or ‘S6’ potassium-binding site, and is indeed likely to stabilize a K^+ ion in this position. We have added Supplementary Figure 7 showing a superposition of emodepside-bound Slo with a structure of the KscA channel under high K^+ conditions where the S6 site is occupied by a hydrated K^+ ion (PDB 1K4C; Zhou et al., 2001). Intriguingly, the hydrated K^+ ion in KscA is located exactly where we observe one of the weak densities which we have modelled as water molecules, and four of the coordinating water molecules overlap with carbonyl oxygen atoms of emodepside. Hence, emodepside appears to stabilize an ion in the S6 site and might assist the removal of the hydration shell. We now also describe this observation in the main text of the revised manuscript.

Furthermore, we have superposed the same KscA structure with verruculogen-bound Slo and found that the hydrated K^+ ion in the S6 position is found just below the four verruculogen molecules. In the new Supplementary Fig. 5F,G, it is obvious that translocating a fully hydrated K^+ ion across the hydrophobic barrier composed of the verruculogen isobutylene moieties would be unfavorable.

3. The hypothesis that emodepside reduces the current through BK channels by creating a sub-optimal potassium binding site below the SF is interesting, especially given that similar mechanism have been proposed for BL (NCA) compounds to actually enhance the current through BK channels (see Schewe et al., Science 2019). It would be of interest if the authors could discuss similarities and differences between these two class of compounds.

We agree with the reviewer that a discussion of similarities and differences of the modes of interaction of verruculogen and emodepside with NCA compounds is very interesting since the binding sites are partially overlapping. We have added a paragraph in the emodepside chapter in the revised manuscript.

Reviewer #3 (Remarks to the Author):

This article by Raisch et al. reported four cryoEM structures of the Drosophila Slo channel in various functional states. These structures revealed potential insect-specific binding pockets and the binding modes of two small molecules, verruculogen and emodepside. Based on these observations, the authors proposed mechanisms for verruculogen and emodepside inhibition. To further support their hypotheses, the authors performed MD simulations to investigate the effect of verruculogen on ion permeation. The cryoEM analyses seem solid and the resolutions of reported structures are sufficiently high for the structural interpretation. The proposed models of verruculogen and emodepside are interesting. This work could be a useful addition to the ion channel field and I would recommend its publication if the following points are addressed or discussed.

Major

1. The effects of emodepside on Ca²⁺ and voltage sensing are very interesting. The abstract made me think that the authors have found the underlying mechanisms and I was disappointed to see that the emodepside-bound structure failed to explain how that happens. Determining structures to explain voltage sensing is difficult and beyond the scope of this study. On the other hand, determining an emodepside-bound structure in the absence of Ca can potentially provide more insights into Ca²⁺ sensing part. But I also understand that is a lot of work. If getting another structure is challenging, the author should at least explicitly discuss the limitation of this study regarding emodepside's modulation on Ca²⁺ and voltage sensing.

We agree with the reviewer that additional structures to elucidate the mechanism of emodepside to uncouple ion translocation from Ca²⁺ and voltage sensing would be very interesting. However, also in agreement with the reviewer, we feel that this would go beyond the scope of the current study and be rather addressed by future follow-up studies. Therefore, as suggested by the reviewer, we have modified the discussion of the emodepside mechanism in the final chapter of the main text to more explicitly mention the limitations of our study.

2. Since the proposed mechanisms of verruculogen and emodepside involve ions in the filter, please show the cryoEM density of K⁺ ion and H₂O (if visible, like those in Supplementary Fig .6G) in the filter and prechamber, with surrounding amino acids contoured at the same level is possible.

We agree with the reviewer that a visualization of the Coulomb potential map around the selectivity filter and the pre-chamber/central cavity would help to evaluate the positions and occupancies of ions, and have added this visualization in Supplementary Figures 9-12.

Minor

1. In the abstract, emodepside is described first and then verruculogen (“Emodepside locks the transmembrane domain.... and voltage sensing. Verruculogen binding inhibits....”), but in the main text, the story of verruculogen is described first. Is there a reason to reverse this order in the abstract?

We thank the reviewer for pointing this out, and have adjusted the abstract so that it follows the order of the main text.

2. Abstract. “Emodepside locks the transmembrane domain in an active conformation...uncoupling ion gating from Ca^{2+} and voltage sensing”. Since the uncoupling effect has already been reported before (Crisford et al., 2015) and the fact that from the current study, it is difficult to understand how the drugs uncouple ion gating from Ca^{2+} and voltage sensing, I suggest revise these sentences so the reader wouldn't have wrong expectations on what questions are answered by this study.

We agree and have modified the abstract accordingly.

3. The authors choose to present the channel upside down (intracellular domain on the top). This is in contrast to what most people in the field do and I do not see any benefit to present Slo channel in this way. But I will let the authors decide whether they want to follow the convention.

As described in the answer to Reviewer #2, we have adjusted all structural figures where this is applicable to make the representation consistent with the literature.

4. Page 6. “...and hand them down to a position between emodepside and the K^+ in position 1 of the filter.”. The position the authors referred to here should be position 4, not position 1. Position 1 is the one near the extracellular side (Zhou et al., 2001).

We have adjusted this in the manuscript.

5. Does emodepside change the ion selectivity? From Supplementary Fig .6B it is hard to see where the reversal potential of the control group is. Although the fact that emodepside does not affect the filter structure suggests the drug does not change ion selectivity, but it would be interesting to know.

We thank the reviewer for raising this interesting question. We agree that, since emodepside does not induce structural changes in the selectivity filter, no drastic change in ion selectivity is expected. The reversal potential is almost identical in absence and presence of emodepside and very close to the potassium equilibrium potential. A slight limitation to this observation, though, is that in our external and internal solutions we use only K^+ and Na^+ as monovalent ions (and not the larger Rb^+). Hence, while it is very clear that emodepside reduces the conductivity under high calcium conditions, it would be very interesting in future to look into ion selectivity in more detail.

REVIEWER COMMENTS

Reviewer #2 (Remarks to the Author):

All my points have been addressed, I therefore recommend the ms for publication. I'd like to congratulate all the authors for very nice science and story.